# Anti-Clogging Performance Optimization for Shunt-Hedging Drip Irrigation Emitters Based on Water–Sand Motion Characteristics



Cheng Qin [1,2,3], Jinzhu Zhang [1,2,3,*], Zhenhua Wang [1,2,3,*], Desheng Lyu [1,2,3], Ningning Liu [1,2,3], Shaobo Xing [1,2,3] and Fei Wang [1,2,3]

1   College of Water Conservancy & Architectural Engineering, Shihezi University, Shihezi 832000, China
2   Key Laboratory of Modern Water-Saving Irrigation of Xinjiang Production & Construction Group, Shihezi 832000, China
3   Key Laboratory of Northwest Oasis Water-Saving Agriculture, Ministry of Agriculture and Rural Affairs, Shihezi 832000, China
*   Correspondence: xjshzzjz@shzu.edu.cn (J.Z.); wzh2002027@163.com (Z.W.)

**Abstract:** To improve the irrigation quality and anti-clogging performance of the emitter, it is necessary to design and optimize its flow channel structure. The shunt-hedging drip irrigation emitter (SHDIE) flow channel is a new type of flow channel. Using computational fluid dynamics, by setting different conditions (such as particle size and injection position), the motion trajectory of sand particles and flow field distribution characteristics of the shunt-hedging flow channel were simulated. According to the simulation results, a new anti-clogging structural optimization scheme was proposed, and physical experiments verified its feasibility. The results showed that the flow index of the original flow channel (SHDIE1) and optimized flow channel (SHDIE2) were 0.479 and 0.486, respectively, which mainly relied on the shunting and hedging of water flow to energy dissipation. For sand particles with diameters of 0.05, 0.10, and 0.15 mm, the average values of the velocity amplitude ratio, $\eta$, were 0.9998, 0.9994, and 0.9991, respectively; the average values of the velocity phase difference, $\beta$, were $-0.143°$, $-0.320°$, and $-0.409°$, respectively. A larger sand particle diameter led to worse followability and a higher risk of blocking the channel. When the sand particles collided with the sensitive region of the flow channel, their movement direction would suddenly change, entering the vortex area. After colliding with the sensitive region of edge A, the maximum probability of sand particles entering the vortex area was increased to 87.5%, and then they stayed in the vortex area under the effect of the sensitive regions of edges B and C. After the sensitive regions were removed, the motion trajectories of sand particles became regular and smooth. The optimized flow channel's (SHDIE2) anti-clogging performance was greatly improved by 60%, with a 1.46% loss of hydraulic performance. This study can provide theoretical support for designing the high anti-clogging emitter.

**Keywords:** shunt-hedging drip irrigation emitters; hydraulic performance; anti-clogging performance; followability; structural optimization

## 1. Introduction

Drip irrigation is one of the most advanced agricultural water-saving irrigation methods, which changes the pressure water flow into droplet flow through the energy dissipation of the emitter. The hydraulic performance of the emitter determines whether the droplet flow is uniform and stable, as well as the irrigation quality of the drip irrigation system [1,2]. The flow index is one of the key parameters used to evaluate the hydraulic performance of drip emitters, and a smaller flow index denotes better hydraulic performance [3,4]. Many research results have shown that the flow channel structure type directly affects the flow index of the emitter. In recent years, researchers have proposed new design ideas for flow channel structure to improve the emitters' hydraulic performance. Li et al. [5] proposed



the fractal flow channel based on the fractal theory, and its flow index was between 0.49 and 0.53. Tian et al. [6] put forward the bidirectional flow channel, whose flow index was less than 0.5. Xing et al. [7] designed the perforated channel according to the principle of bionics with a minimum flow index of 0.46.

In practice, many tiny sand particles are carried in the irrigation water. Before entering the emitter, most sand particles can be intercepted through various engineering measures but not completely removed [8–10]. Therefore, in addition to the requirement for excellent hydraulic performance, the emitter needs to be able to prevent or reduce flow channel blockages caused by sediment particles to prolong the service life of the drip irrigation system [11,12]. Optimizing the emitter's flow channel is the most direct and effective way to relieve the blockage issue. Wei et al. [13] improved the anti-clogging performance of the emitter by removing the low-speed vortex area in the flow channel. Niu et al. [14] based on the simulation results of the two-phase flow model, selected the smaller isoline of sediment concentration as the new boundary of the flow channel to reduce the blockage caused by waterborne particles. Yang et al. [15] increased the turbulence degree of the flow field by eliminating the area with small turbulent kinetic energy in the flow channel to improve the anti-clogging performance of the emitter. However, the above optimization scheme cannot fully consider the loss of the hydraulic performance of the emitters, or the boundary shape of the optimized flow channel is too complex, which is not conducive to manufacturing and cost control.

One approach is to make the sand particles pass through all flow channel units smoothly instead of aggregated in the vortex area. In that case, the anti-clogging performance of the emitter would be significantly improved. At present, most of the structural optimization studies have focused on the optimization of the flow field in the flow channel, whereas reports on the optimization focused on the sand particle's motion characteristics in the flow channel are scarce in the literature [13–15]. Taking the original flow channel before optimization (SHDIE1) and the new flow channel after optimization (SHDIE2) of shunt-hedging drip irrigation emitters (SHDIEs) as the research object, the movement characteristics of water–sand in the flow channel were studied through numerical simulation. A new structural optimization scheme is proposed according to the numerical simulation results, and physical experiments verify its feasibility. The relevant conclusions can provide theoretical guidance and technical support for the optimal design of the emitter.

## 2. Materials and Methods

### 2.1. Physical Model

The energy dissipation modes of the shunt-hedging flow channel include the sudden expansion of the flow channel and the shunt-hedging of the water flow. The structural parameters of the shunt-hedging flow channel are shown in Figure 1. There were eight flow channel units in the shunt-hedging flow channel, with a depth of 0.8 mm and a width of 2.6 mm. In order to improve the velocity of water near the wall and the self-cleaning ability of the flow passage, some structural shapes were designed as arcs [8,16]. Each flow channel unit included a shunt part like the letter D and two symmetrically distributed arc-shaped diversion parts (hereinafter referred to as shunt parts and diversion parts), whose functions were to shunt the water flow and guide hedging, respectively.

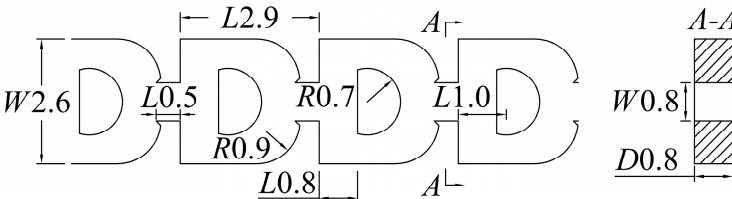

**Figure 1.** The parameters and test pieces of shunt-hedging emitter.

### 2.2. Field Test

As shown in Figure 2, a comprehensive performance test platform was used for clear water and short-cycle anti-clogging tests, mainly composed of a water tank, water pump (WQS-26-1.5, pump rated head = 26 m, pump maximum head = 40 m), one pressure gauge (YB-150B, measurement range = 0.00–0.16 MPa, accuracy level = 0.25), four valves, and several PE pipelines. During the experiment, for emitters with different structures, three pieces were taken at the same time to test the flow rate. All test pieces were printed using a 3D printer (nanoArch S140 Micro nano 3D printer, BMF Precision Tech Inc, Chongqing, China), and the printing accuracy was 0.01 mm, as measured by an electron microscope (Dino-Lite Polarizer, AnMo Electronics Corporation, Taiwan, China). The pressure range of the clear water test was consistent with the inlet pressure of the numerical simulation. Each test was conducted for 5 min, and the average of the three test pieces was taken as the measured flow. The annual average sediment concentration in the Manas Basin of Shihezi Irrigation District in Xinjiang, China, was 3 kg/m$^3$ [17]. Therefore, the muddy water with a sand concentration of 3 g/L was configured for the anti-clogging test, and the sand in the test was taken from the riverbed of Manas River, whose diameter was less than 0.15 mm. The flow rate was tested according to the periodic intermittent irrigation methods. Under the pressure of 0.10 MPa, each anti-clogging test lasted for 30 min, and the flow rate was calculated by weighing the weight of the effluent in the last 5 min. The above steps were repeated until the emitter's flow rate decreased to less than 75% of the clear water flow.

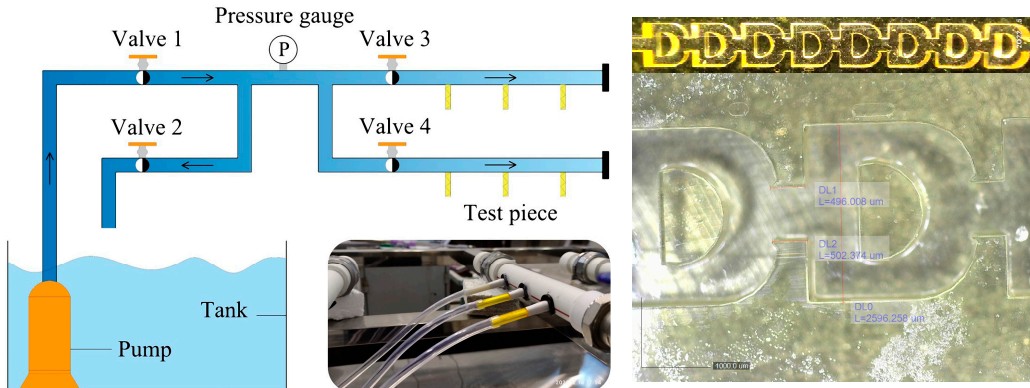

**Figure 2.** Experiment platform and test pieces.

### 2.3. Numerical Methods

The geometry model of the emitter was established by NX 12.0, meshed by ICEM CFD, and calculated by Fluent 19.0. The mesh independence test was carried out to ensure the calculation accuracy and reduce the cost of the processor. As shown in Figure 3a, when the number of grids was more than 940,000, the emitter's flow change rate was low, indicating that the flow value was not sensitive to the change of the grid's number. Finally, the model with 940,000 grids and a maximum grid size of 0.036 mm was selected for numerical simulation (Figure 3b).

The narrow and tortuous flow channel inside the emitter puts the fluid in a turbulent state, which meets the requirements of the mathematical fluid model in the flow channel established by the Navier–Stokes equation [4]. Using FLUENT software for simulation calculation, studies have shown that the standard *k–e* model can accurately reflect the actual flow when applied to the flow field calculation of the emitters [4,18]. In addition, using the standard wall function defaulted by Fluent software to process all walls can make the calculation results of *k–e* model more accurate [19]. Therefore, the standard *k–e* model and standard wall function were selected for simulation calculation. The inlet pressure range was 0.05–0.20 MPa with an interval of 0.01 MPa, and the outlet pressure was always 0. To accelerate the convergence process and ensure good convergence, the "SIMPLEC" algorithm with less relaxation was selected to couple velocity and pressure [20]. The second-order upwind method was used to solve the convection term, and the convergence accuracy standard was set to $10^{-5}$. With

an inlet pressure set to 0.10 MPa, a discrete phase model was used to simulate and analyze a single sand particle's trajectory and velocity distribution in the channel. Six injection positions were set for each particle at the inlet to make the simulation results more reliable, as shown in Figure 3c. The particle density was set to 2500 kg/m$^3$, with diameters of 0.05, 0.10, and 0.15 mm. In addition, a stochastic particle trajectory model was applied to the simulation, and the effects of gravity, buoyancy, and virtual mass force were considered [21]. As the fluctuation of a turbulent fluid causes particle diffusion, the fluid was regarded as the continuous phase and the sand was regarded as the discrete phase, and a bidirectional coupling calculation was adopted [15].

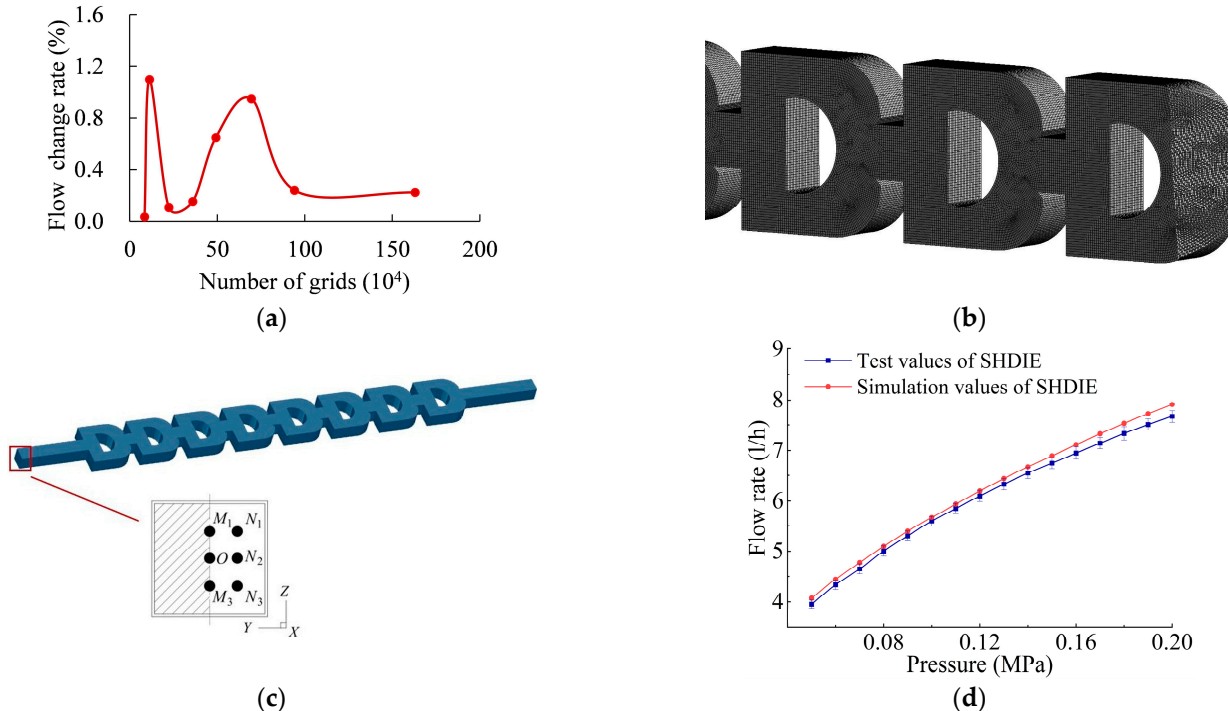

**Figure 3.** Meshing and parameter setting: (**a**) mesh independence test; (**b**) fluid domain meshing; (**c**) inject position of sand particles; (**d**) pressure flow curves of SHDIE1.

Figure 3d shows the pressure–flow curve of SHDIE1. The flow rate error between the simulated value and the test value of SHDIE1 was 1.29–3.21%, which means that the numerical simulation can accurately reflect the flow field of the flow channel.

### 2.4. Evaluation Indicators

The value of the flow index is between 0 and 1, which reflects the sensitivity of flow to pressure. A smaller value denotes a better hydraulic performance of the emitters. The flow index equation is as follows:

$$q = k(H \times 10^{-3})^x, \tag{1}$$

where $q$ is the channel's average flow rate (L/h), $k$ is the flow coefficient, $x$ is the flow index, and $H$ is the inlet pressure (MPa).

Under the same pressure, the ratio of the muddy water flow to the clear water flow of the emitter is called the relative flow rate, which is recorded as $q_r$. When $q_r < 75\%$, the emitter is considered completely blocked [22]. The relative flow equation is as follows:

$$q_r = \frac{q_i}{q_0} \times 100\%, \tag{2}$$

where $q_i$ is the muddy water flow of the emitter (L/h), $i$ is the number of muddy water tests, and $q_0$ is the clear water flow of the emitter under the same inlet pressure (L/h).

The formula for particle followability can be deduced using the Fourier integral of fluid and particle velocity [23]. The particle's following behavior to fluid movement in large numbers of solid–liquid two-phase flow problems in hydraulic engineering, environmental engineering, and many other fields can be characterized by the amplitude ratio and phase difference between particle velocity and fluid velocity [24]. The velocity amplitude ratio, $\eta$, and the velocity phase difference, $\beta$, are general parameters for evaluating particle followability. The followability calculation of sand particles was based on the simulation results of the DPM model. After the DPM model completes the simulation of sand particle motion, the angular frequency, $\omega$, of fluid motion around sand particles can be obtained through Fluent software. Finally, the density and diameter of sand particles are substituted under different simulation conditions into Equations (3)–(8) to calculate the values of $\eta$ and $\beta$. Values of $\eta = 1$ and $\beta = 0°$ indicate that the particles move completely with the fluid; $\eta < 1$ and $\beta < 0°$ indicate that the particles lag behind the fluid motion; $\eta > 1$ and $\beta > 0°$ indicate that the particles move ahead of the fluid. The calculation formulas for particle followability are as follows:

$$\eta = \sqrt{\left(1+f_1\right)^2+f_2^2}, \tag{3}$$

$$\beta = \tan^{-1}\left[f_2/\left(1+f_1\right)\right], \tag{4}$$

where

$$f_1 = \frac{\left[1+\frac{9N_s}{\sqrt{2}(s+1/2)}\right]^{\frac{1-s}{s+1/2}}}{\frac{81}{(s+1/2)^2}\left(2N_s^2+\frac{N_s}{\sqrt{2}}\right)^2+\left[1+\frac{9N_s}{\sqrt{2}(s+1/2)}\right]^2}, \tag{5}$$

$$f_2 = \frac{\frac{9(1-s)}{(s+1/2)^2}\left(2N_s^2+\frac{N_s}{\sqrt{2}}\right)}{\frac{81}{(s+1/2)^2}\left(2N_s^2+\frac{N_s}{\sqrt{2}}\right)^2+\left[1+\frac{9N_s}{\sqrt{2}(s+1/2)}\right]^2}, \tag{6}$$

$$N_s = \sqrt{\nu/\omega d_p^2}, \tag{7}$$

$$s = \rho_p/\rho_f, \tag{8}$$

where $N_s$ is the Stokes number, $\nu$ is the kinematic viscosity coefficient of the fluid, $\omega$ is the angular frequency of moving fluid, $d_p$ is the particle diameter (mm), $s$ is the density ratio of particle to fluid, $\rho_f$ is the density of particles (g/cm$^3$), $\rho_f$ is the density of the fluid (g/cm$^3$), and $f_1$ and $f_2$ are the pulsating frequencies.

## 3. Results

### 3.1. Hydraulic Characteristics of SHDIE1

Using Equation (1) to fit the curve in Figure 3d, the flow index of SHDIE was 0.479, and the coefficient of correlation was between 0.998 and 0.999. The pressure nephogram shows that the pressure decreased gradually along the flow direction. After the water flowed through the shunt part and the hedging area, the pressure dropped observably, which indicates that shunt and hedging were the main energy dissipation methods. Figure 4 shows the velocity distribution of the flow channel, and each flow channel unit had a similar flow field distribution. The flow field was divided into three regions according to the fourth channel unit's velocity nephogram and streamline diagram. The vortex area on the back of the diversion parts was defined as area I, and the vortex area on the back of the shunt parts was defined as area II. The remaining area was defined as mainstream area III, and the water flow hedging area was defined as area III*. The velocity range of mainstream area III was 1.21–4.53 m/s, and the velocity range of vortex areas I and II was 0.11–1.21 m/s. Figure 5 shows the movement of sand particles in different areas of the flow channel, and the velocity of sand particles in the vortex and mainstream areas was different. The velocity of sand particles in the vortex area was low, and the velocity range was 0.06–1.10 m/s. This shows that the probability of sand particle deposition is high after entering the vortex area from the mainstream area. However, the vortex area II is narrow and long, and the arc boundary

is not conducive to the stable aggregation of sand particles. In comparison, the vortex area I surrounded by the right-angle boundaries not only promotes the stable deposition of sand particles, but the sand particles also adhere to each other to form aggregates under the action of the large eddy, resulting in a high risk of flow channel blockage. This is consistent with the results of the muddy water test, as shown in Figure 6. SHDIE was completely blocked after the 15th test due to the continuous aggregation of sand particles in vortex area I.

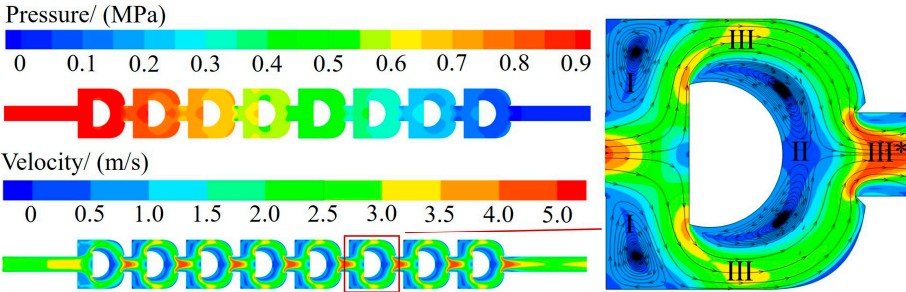

**Figure 4.** Velocity and pressure contour of SHDIE2. * shows the pressure and velocity distribution of the internal flow channel of the shunt-hedging drip emitter, and divides the flow channel's areas according to the velocity distribution of the sixth flow channel unit.

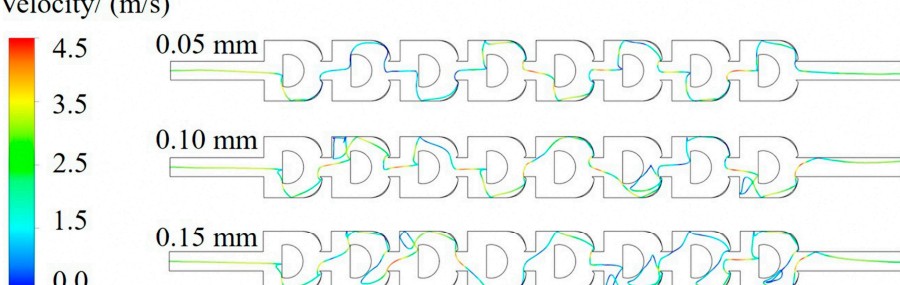

**Figure 5.** The trajectory of the inject sand particles at *O*.

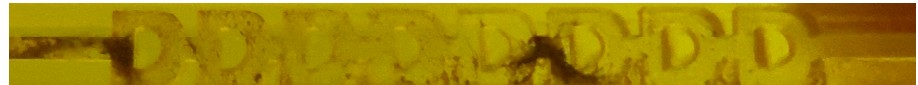

**Figure 6.** SHDIE1 blocked at the 24th cycle.

### 3.2. Motion Characteristics of Sand Particles

For the sand particle with a diameter of 0.05 mm, its motion trajectory was smooth and regular, as shown in Figure 7a. Its velocity coincided with the surrounding fluid velocity, and the average velocity was 2.25 m/s. As shown in Figure 7b, for the sand particle with a diameter of 0.10 mm, its average velocity was 1.90 m/s, and the deviation between the sand particle velocity and the surrounding fluid velocity increased, whose motion trajectory became confused. As shown in Figure 7c, for the sand particle with a diameter of 0.15 mm, its average velocity was 1.56 m/s, and there was a significant deviation between the sand particle velocity and the surrounding fluid velocity, whose motion trajectory was disordered. This is because, with the increase in diameter, the resistance borne by the sand particles increased during movement, and water's carrying effect on sand particles decreased, worsening the sand particle's followability [21]. After the large sand particle collided with the boundary of the flow channel, it deviated from the mainstream in some areas and entered the vortex areas for irregular motion. Therefore, compared with the tiny sand particles that can follow the mainstream well and directly flow out of the flow channel. The motion trajectories of 0.10 and 0.15 mm sand particles were significantly irregular, and the transportation distance was longer. Table 1 shows the transportation distance of sand particles in the flow channel at different injection positions. The average transportation

distance of the sand particles with diameters of 0.05, 0.10, and 0.15 mm in the flow channel was 44.78, 56.32, and 62.87 mm, respectively. Furthermore, the statistical results showed that the total number of times the sand particles of 0.05, 0.10, and 0.15 mm entered the vortex area was three, 18, and 24, respectively. From this, the movement characteristics of sand particles incident at different positions were similar. With the increase in sand particle diameter, the distance of sand particles moving in the vortex area increased.

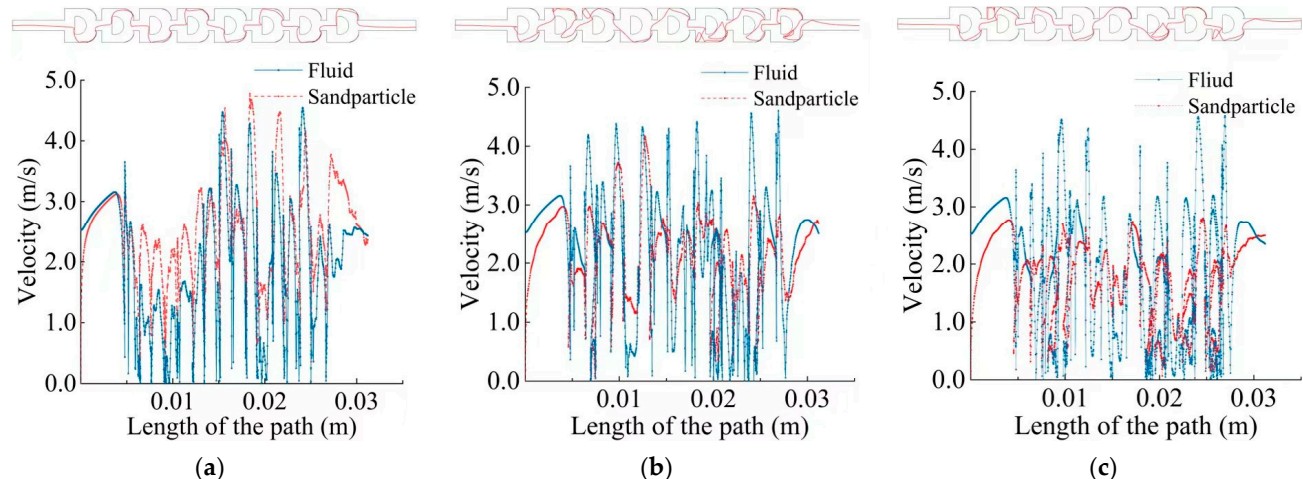

**Figure 7.** The trajectory of the inject sand particles at *O* and the velocity change between sand particles and surrounding fluid: (**a**) 0.05 mm; (**b**) 0.10 mm; (**c**) 0.15 mm.

**Table 1.** Transportation distance of sand particles in the flow channel at different injection positions.

| Diameter (mm) | Transportation Distance (mm) | | | | | | |
|---|---|---|---|---|---|---|---|
| | $M_1$ | $O$ | $M_2$ | $N_1$ | $N_2$ | $N_3$ | Average |
| 0.05 | 52.46 | 45.57 | 49.11 | 47.18 | 46.37 | 45.61 | 47.72 |
| 0.10 | 62.58 | 54.96 | 59.43 | 56.94 | 47.24 | 56.80 | 56.32 |
| 0.15 | 63.19 | 62.26 | 58.18 | 67.58 | 62.82 | 63.32 | 62.89 |

In order to further analyze the followability and motion characteristics of sand particles with different diameters, sand particles injected at position *O* were taken as an example. The followability of particles was used to evaluate the blockage possibility to the flow channel. As $\eta$ approached 1 and $\beta$ approached 0°, the sand particles better followed the water flow [24]. As shown in Figure 8, the average values of $\eta$ and $\beta$ decreased with the increase in sand particle diameter. For sand particles with diameters of 0.05, 0.10, and 0.15 mm, the average values of $\eta$ were 0.9998, 0.9994, and 0.9991, respectively, and the average values of $\beta$ were −0.143°, −0.320°, and −0.409°, respectively, indicating that the sand particle size had a significant influence on its movement. A larger sand particle diameter led to worse followability. This is consistent with the motion trajectory and velocity variation characteristics of sand particles shown in Figure 6. The sand particles with poor followability would move for a longer distance and time in the flow channel, and they would be easy to deposit [15].

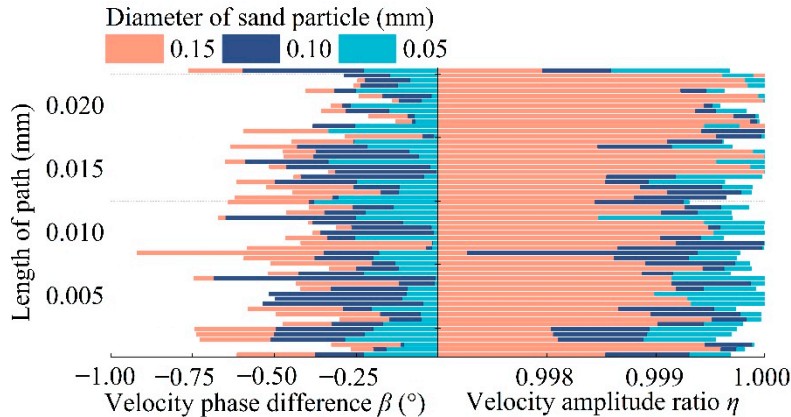

**Figure 8.** Velocity amplitude ratio and velocity phase difference of sand particles inject at *O*.

According to the motion characteristics of the sand particles, there were some sensitive edges in the emitter channel, and the probability of the sand particles colliding with them into the vortex area was greater, thus increasing the risk of the flow channel blockage. The upstream surface of the shunt parts was defined as edge A. The boundaries of the vortex area I were defined as edges B and C, respectively. Sensitive regions were divided according to the farthest collision point, which caused sand particles to enter the vortex area I. The extreme values of the collision points were $L_{A, max}$ = 0.58 mm, $L_{B, max}$ = 0.64 mm, and $L_{C, max}$ = 0.9 mm, respectively. Therefore, the sensitive regions of edges A, B, and C were divided into 0–0.58, 0–0.64, and 0–0.90 mm, respectively. The probability of the sand particles entering vortex area I after collision with edges A, B, and C under different injection positions with different particle diameters was counted, as shown in Table 2. After the sand particles collided with the sensitive region of edge A, the probabilities of the sand particles with diameters of 0.05, 0.10, and 0.15 mm entering vortex area I were 9.1%, 72.2%, and 87.5%, respectively. The sand particles that collided with the sensitive regions of edges B and C were sure to enter the vortex area I once again. The results showed that the sensitive region on the edges of the flow channel significantly increased the times and probability of sand particles entering the low-speed vortex area. In the low-speed vortex area, the water's carrying effect on sand particles was poor, thus significantly increasing the probability of sand deposition [15].

**Table 2.** The collision situation between sand particles with different sizes and the boundary, and the probability of entering the vortex area I.

| Collision Situation | | | | |
|---|---|---|---|---|
| 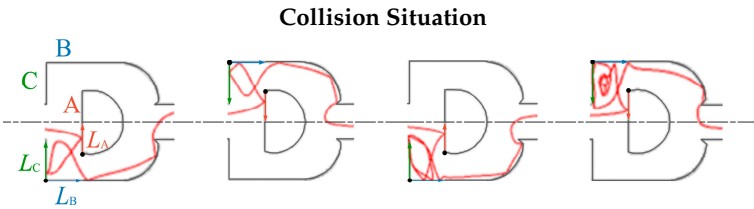 | | | | |

| Diameter (mm) | Probability | | | | |
|---|---|---|---|---|---|
| | $0 \leq L_A \leq 0.58$ mm | $0.58 \leq L_A$ mm | $0 \leq L_B \leq 0.64$ mm | $0.64 \leq L_B$ mm | $0 \leq L_C \leq 0.90$ mm |
| 50 | 9.1 | 0 | 100 | 0 | 100 |
| 100 | 72.2 | 0 | 100 | 0 | 100 |
| 150 | 87.5 | 0 | 100 | 0 | 100 |

### 3.3. Optimization of the Flow Channel Structure

Considering the flow channel blockage caused by the sensitive edges and the ease of manufacturing, linear geometry was used for the structural optimization of SHDIE1, as shown in Figure 9a. On the one hand, the sensitive region of edge A was removed to prevent the sand particles from entering vortex area I after collision with the sensitive region. If the whole sensitive region on edge A was completely removed radially, this would destroy the flow channel's basic functions of shunt and energy dissipation. Therefore, the average value of all collision points between sand particles and the sensitive region of wall A was calculated to be $L_A = 0.39$ mm, and the region to be optimized on edge A was determined as 0–0.39 mm. The cutting angle $\theta$ was consistent with the direction of the local velocity vector of water flow at $L_A = 0.39$ mm, i.e., $\theta = 41°$, to ensure that the shape of the optimized boundary was consistent with the movement direction of the sand particles and adapted to their subsequent motion direction. On the other hand, due to the influence of the sensitive regions of edges B and C, the escape of sand particles entering the vortex area I was challenging. Therefore, the low-speed vortex area covered by the right triangle composed of the sensitive regions of edges B and C was filled. The optimized emitter was recorded as SHDIE2.

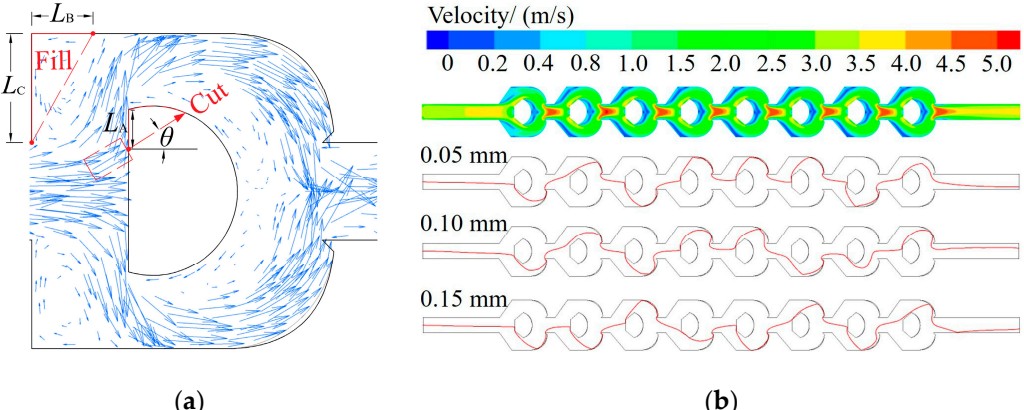

**(a)** **(b)**

**Figure 9.** Optimization results: (**a**) schematic diagram of structure optimization; (**b**) the velocity nephogram of SHDIE2 and motion trajectory of inject sand particles at *O*.

Figure 9b shows the velocity distribution and the sand particle's trajectory in SHDIE2. The maximum flow velocity of SHDIE2 reached 5 m/s, and the proportion of the mainstream area was 21% higher than that of SHDIE1. In addition, sand particles migrated more smoothly in SHDIE2. The average transportation distances of the sand particles with diameters of 0.05, 0.10, and 0.15 mm decreased to 44.78, 43.28, and 42.65 mm, respectively, and the total number of times the sand particles entering the vortex area decreased to three, two, and two, respectively. Through structural optimization, the sand particles could flow better in the mainstream area, significantly improving the emitter's anti-clogging performance. In addition, a clear water and anti-clogging test for SHDIE2 was carried out. Figure 10a is the hydraulic characteristic curve of SHDIE1 and SHDIE2. The flow index of the SHDIE2 was 0.486, and hydraulic performance decreased by 1.5%. However, the anti-clogging performance of SHDIE2 was significantly improved. Figure 10b,c show that SHDIE2 was wholly blocked in the 24th short-cycle anti-clogging test, while SHDIE1 was completely blocked in the 15th test. Structural optimization could greatly improve the anti-clogging performance of the shunt-hedging flow channel by 60%.

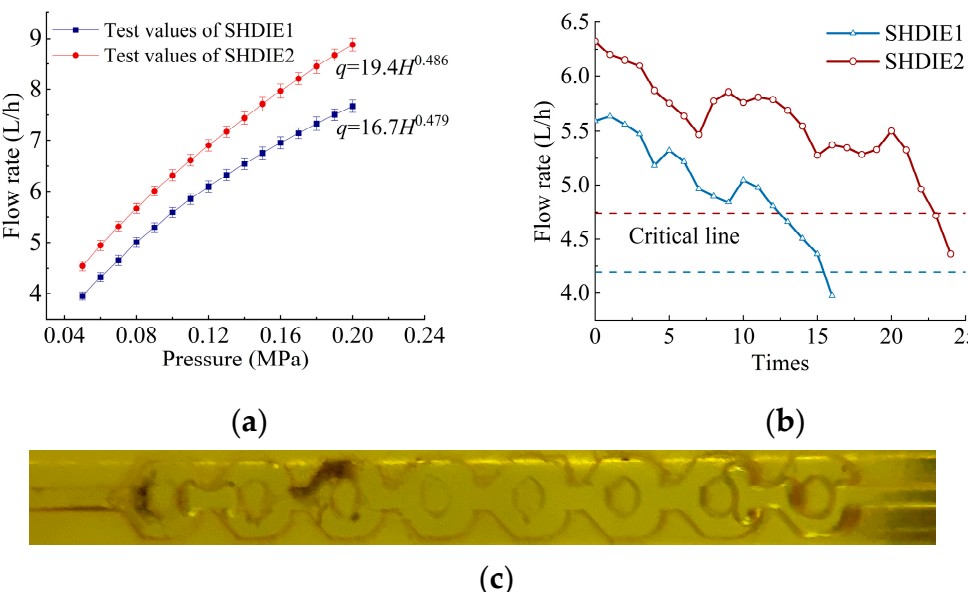

**Figure 10.** Physical test results: (**a**) hydraulic characteristic curves; (**b**) short-cycle anti-clogging test results; (**c**) SHDIE2 was blocked at the 24th time.

## 4. Discussion

Local head loss is the primary pathway of energy dissipation and can effectively improve the hydraulic performance of emitters [25]. The shunt-hedging drip irrigation emitter generates local head loss by shunting and hedging the water flow and the sudden expansion of the channel. The relative errors between the numerical simulation results and the test values were 1.29–3.21%, indicating that the data obtained using CFD software in this paper were reliable. It can be found that the test value was less than the simulated value, which is consistent with other studies [18]. The main reason is the additional energy dissipation caused by the roughness of the specimen wall.

The pressure nephogram of SHDIE1 showed that the pressure decreased significantly after the water flowed through area III*, which indicates that the shunt and hedge were the main forms of energy dissipation of SHDIE1. From the velocity nephogram of SHDIE1, it can be seen that there were vortex areas I and II in the flow field. The existence of the right-angle boundary in the vortex area I made the sand particles aggregate stably and resulted in a high risk of flow channel blockage. To verify this point, the motion of sand particles with different diameters in SHDIE1 was simulated. $\eta$ and $\beta$ were used to quantitatively analyze the followability of sand particles. A larger diameter of the sand particle resulted in lower average values of $\eta$ and $\beta$, as well as worse followability. The movement characteristics of the sand particles confirmed the previous view. With the particle diameter increasing, the sand particles' motion trajectory became more disordered, and the velocity deviation between the sand particles and the surrounding fluid increased. Firstly, a larger particle size of sand led to a more disordered motion trajectory. Secondly, a larger particle size of sand led to a larger velocity deviation between the sand and the surrounding fluid. As also indicated in the studies of Yu et al. [26] and Tang et al. [21], the reason is that larger sand particles have greater resistance during movement, resulting in a poor ability to be wrapped and carried by water flow. When large sand particles collide with the sensitive region of the boundary, they can easily change the motion direction and enter vortex area I. In addition, the physical test results also showed that the sand particles were mainly aggregated in vortex area I, further validating the above viewpoint.

SHDIE2 was obtained by eliminating the sensitive regions of SHDIE1. The mainstream area of the optimized flow channel increased by 21%, and the sand particles in the flow channel could pass through all flow channel units smoothly. Since the average water velocity in the SHDIE2 channel was higher and the maximum velocity could reach 5 m/s, the optimized

flow channel had a better "self-cleaning" effect [27]. The results of numerical simulation and physical tests showed that the anti-clogging performance of SHDIE2 was greatly improved by 60%, while the flow index of SHDIE2 was only 1.46% lower than that of SHDIE1. One of the main reasons is that only the sensitive edges of the flow channel were optimized to promote the smooth movement of sand particles. Most of the boundaries of the flow channel were unchanged, and the flow field changed a little before and after the channel optimization, which better coordinated the contradiction between hydraulic performance and anti-clogging performance. In addition, removing the vortex area could prevent sand particles from entering the vortex area and improve the anti-clogging performance of the flow channel [26]. However, Feng et al. [28]. pointed out that the vortex area can dissipate energy through friction with the side wall. When the vortex area was eliminated, the hydraulic performance naturally decreased.

This work proposed an anti-clogging optimization scheme and a shunt-hedging drip irrigation emitter. However, the motion analysis of sand particles and the structural optimization of the flow channel in this study were all based on simulation data. In the future, particle image velocimetry technology can be used to study the actual motion characteristics of water–sand to achieve more accurate structure optimization.

## 5. Conclusions

In this work, a new flow channel structure of the emitter was designed. The motion characteristics of water–sand in the channel were studied, and a channel structure optimization method based on the sand particle's motion characteristics was proposed. The conclusions are as follows:

(1) The flow index of the shunt-hedging drip irrigation emitter was about 0.479. The hydraulic performance of SHDIE was excellent, mainly depending on hedging the water flow to dissipate energy.

(2) A larger sand particle had worse followability. After the collision with the sensitive region of the flow channel structure, it could easily enter the vortex area and aggregation, which was the main reason for the emitter's blockage. In the pretreatment of irrigation water sources, attention should be paid to removing large impurities above 0.10 mm.

(3) The flow channel structure optimization method based on the motion characteristics of sand particles could significantly improve the anti-clogging performance while only losing little hydraulic performance. In this paper, the flow index of the optimized emitter only decreased by 1.46%, while its anti-clogging performance greatly improved by 60%.

**Author Contributions:** J.Z. and Z.W. provided the research methods and writing ideas of the manuscript; C.Q. conducted data analysis and wrote the original draft; D.L. and N.L. provided important advice on concepts; S.X. and F.W. reviewed the manuscript. All authors have read and agreed to the published version of the manuscript.

**Funding:** This research was funded by the South Xinjiang Key Industry Innovation and Development Support Plan Project of XPCC (2020DB004), the Major Science and Technology Projects of XPCC (2021AA003), the National Natural Science Foundation of China (52279040), the National Key Research and Development Projects (2021YFD19008003), and the National Key Research and Development Projects (2022YFD1900405).

**Institutional Review Board Statement:** Not applicable.

**Informed Consent Statement:** Not applicable.

**Data Availability Statement:** The data that support the finding of this study are available from the corresponding author upon reasonable request.

**Acknowledgments:** We sincerely thank the College of Water Conservancy & Architectural Engineering (Shihezi University) for providing the experimental site.

**Conflicts of Interest:** The authors declare no conflict of interest.

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
