# Peer review of "Anti-Clogging Performance Optimization for Shunt-Hedging Drip Irrigation Emitters Based on Water–Sand Motion Characteristics"

_water, doi:10.3390/w14233901_

Round 1
Reviewer 1 Report
Review: comments and suggestions
Anti-clogging performance optimization for shunt-hedging drip irrigation emitters based on water-sand motion characteristics
Abstract
Is well articulated except for a slight modification
Introduction
It is good but there are some suggestions to improve.
Methods
Needs to include the measurements or data input used in the test. The authors should describe it more in the section.
Results
Section 3.2 Motion characteristics of sand particle
In graphs b and C, velocity motion at 0.01 and 0.15mm looks very similar to the 0.05mm diameter.
Line 191 why did the trajectory pattern become confused? What is the logic behind it? There should be scientific justification for such kinds of results.
There are also additional comments that need explanations.
Discussion part
Is well-written and discussed
Conclusion part
It did not include the limitation and the future perspectives of the study.
It should explain the limitation of the study, what is missed and what could have been included.
What future can be done or improved on it?
General comments
It is possible to accept it with minor improvement.

Author Response
Dear Reviewer:Please see the attachment.

Reviewer 2 Report
The problem of sand clogging has been a complicated and hot issue in drip irrigation in recent years. In this paper, the migration process of sand particles in the emitters was studied by combining experiments and numerical simulation. The emitters were optimized according to the characteristics of the flow field, and satisfactory results were obtained. This study is interesting and has potentially high value. It can provide some reference for the structure optimization of the emitter. Generally speaking, the manuscript is helpful for the readers of the Water journal and can be published. However, there are still some problems in manuscript, meanwhile, the manuscript needs to be checked carefully for grammar and sentence pattern. Some specific comments for improving the manuscript are as follows:
(1) Figure 1 on page 2: There is something wrong with the labeling. For example, the labeling of radius should be "R0.7".
(2) Line 92 to 93 on page 3: Please add the specific model, parameters, and accuracy of each device.
(3) Line 96 on page 3: Please confirm whether the 3D printer can reach the accuracy of 0.01mm.
(4) Line 112 on page 3: Figure3a is a grid diagram only, so Figure3a should be placed after ICEM CFD.
(5) Line 114 on page 3: The sentence "when the number of grids is 940 thousand, the emitter's flow change rate is low" should be expressed as “when the number of grids is more than 940 thousand, the emitter's flow change rate is low”.
(6) Line 118 on page 3: Why choose k- turbulence model instead of DDES or LES?
(7) Line 119 on page 3: The author adopted wall function in the process of numerical simulation. How to consider Y plus?
(8) Line 126 to 128 on page 3: It is an innovative idea for the author to set 6 injection positions, but if so, it is inconsistent with the actual flow conditions. How to ensure the correct numerical results?
(9) Equation 1 on page 4: Please indicate the significance of the k value.
(10) Line 160 to 163 on page 5: Please indicate the physical meaning of f1 and f2.
(11) Line 177 to 185 on page 5: Since the DPM method was adopted in this paper to simulate the motion of sand particles in the emitter, the author should combine the flow field with the sand particle distribution diagram under different particle sizes to make the analysis results more convincing.
(12) Line 197 to 202 on page 6: According to Figure 6, it can be indicated that the tiny particle size, the less velocity difference between the particle and the fluid, and the fewer times it enters the vortex area. Meanwhile, the velocity of the sand diameter of 0.05mm is also the largest. However, why the smaller the particle size is, the shorter the travel distance of the sand particle is? Please give a detailed explanation.
Author Response

(The authors gave the same response as above.)

Round 2
Reviewer 2 Report
After modification, the content is more substantial, and the logic is more precise, which can meet the requirements for publication in the Water Journal. It is recommended to publish.